# Formulation of Magneto-Responsive Hydrogels from Dually Cross-Linked Polysaccharides: Synthesis, Tuning and Evaluation of Rheological Properties

**DOI:** 10.3390/ijms23179633

**Published:** 2022-08-25

**Authors:** Lenka Vítková, Lenka Musilová, Eva Achbergerová, Roman Kolařík, Miroslav Mrlík, Kateřina Korpasová, Leona Mahelová, Zdenka Capáková, Aleš Mráček

**Affiliations:** 1Department of Physics and Materials Engineering, Faculty of Technology, Tomas Bata University in Zlin, Vavrečkova 275, 760 01 Zlin, Czech Republic; 2Centre of Polymer Systems, Tomas Bata University in Zlin, tř. Tomáše Bati 5678, 760 01 Zlin, Czech Republic; 3CEBIA-Tech, Faculty of Applied Informatics, Tomas Bata University in Zlin, Nad Stráněmi 4511, 760 05 Zlin, Czech Republic

**Keywords:** hyaluronan, smart hydrogels, magnetorheology, Schiff base, hydrodynamic radius, tissue engineering

## Abstract

Smart hydrogels based on natural polymers present an opportunity to fabricate responsive scaffolds that provide an immediate and reversible reaction to a given stimulus. Modulation of mechanical characteristics is especially interesting in myocyte cultivation, and can be achieved by magnetically controlled stiffening. Here, hyaluronan hydrogels with carbonyl iron particles as a magnetic filler are prepared in a low-toxicity process. Desired mechanical behaviour is achieved using a combination of two cross-linking routes—dynamic Schiff base linkages and ionic cross-linking. We found that gelation time is greatly affected by polymer chain conformation. This factor can surpass the influence of the number of reactive sites, shortening gelation from 5 h to 20 min. Ionic cross-linking efficiency increased with the number of carboxyl groups and led to the storage modulus reaching 10${}^{3}$ Pa compared to 10${}^{1}$ Pa–10${}^{2}$ Pa for gels cross-linked with only Schiff bases. Furthermore, the ability of magnetic particles to induce significant stiffening of the hydrogel through the magnetorheological effect is confirmed, as a 10${}^{3}$-times higher storage modulus is achieved in an external magnetic field of 842 kA·m${}^{-1}$. Finally, cytotoxicity testing confirms the ability to produce hydrogels that provide over 75% relative cell viability. Therefore, dual cross-linked hyaluronan-based magneto-responsive hydrogels present a potential material for on-demand mechanically tunable scaffolds usable in myocyte cultivation.

## 1. Introduction

Hydrogels, i.e., 3D cross-linked, high-water-content networks, have been used in many biomedical applications [1], e.g., fabrication of scaffolds for cell cultivation [2] and tissue engineering [3], drug delivery [4], and wound dressing [5], as well as industrial applications such as soft electronics [6], water treatment [7,8], food applications [9,10], etc. Each of these applications requires a tailored hydrogel with desired characteristics. Basic properties of 3D networks are given by their chemistry as well as the incorporation of other components. Tissue engineering application in particular require close resemblance to native tissue in terms of morphology, as well as functionality, which often includes sudden and reversible changes of properties. In the case of smart hydrogels, their composition enables them to respond to external stimuli (pH, ionic strength, temperature, or electric or magnetic fields), and thus their properties can be advantageously dynamically changed. Namely, magnetorheological (MR) materials can rapidly and reversibly alter the mechanical properties of hydrogels under a magnetic field due to the magnetorheological effect (MRE) [11].

MRE refers to a highly non-linear mechanical response of a material to the presence of an external magnetic field. This is commonly achieved in multi-phase systems containing magnetically active filler (e.g., iron oxide-based nanoparticles Fe${}_{3}$O${}_{4}$ (magnetite) or γ-Fe${}_{2}$O${}_{3}$ (maghemite) [12]; superparamagnetic iron-oxide particles (SPIONs) [13]; or CoFe${}_{2}$O${}_{4}$ [14]) dispersed in a non-magnetic matrix [15]. Iron-based magnetic fillers are advantageous in biological applications due to their degradability in vivo [16] through a biotransformation mechanism whereby the magnetic material is converted into nontoxic iron species within acidic intracellular lysosomes [17]. Further, carbonyl iron particles have been found to be nontoxic to living cells [18,19], and their hydrogels based on Poly(2-oxazolines) show very promising magneto-responsive and noncytotoxic behaviour [11]. Applying an external magnetic field induces magnetic dipoles in the filler particles, causing the particles to be in order, effectively forming a network and thus leading to material stiffening [20]. This allows on-demand reversible modulation of mechanical properties. The magnitude of MRE is given by the magnetization saturation and the volume fraction of the magnetic filler, and the viscous dissipation of carrier medium [15,21]. Due to viscosity restrictions, MRE is generally greater in liquid carrier media [22]. Nevertheless, significant stiffening has been achieved in solid matrices such as hydrogel [23,24,25] as well. Furthermore, it has been shown that complex rheological tuning of the carrier medium, specifically its yield stress, may produce a solid system with virtually unaffected MR efficiency [26]. In terms of life sciences and tissue engineering, this can be exploited for arterial embolization in cancer treatment [27,28] or mechanical stimulation of cells via an external stimulus [29,30].

Cross-linked hydrogel matrix can be made from both synthetic and natural polymers. In most cases, synthetic polymers present a challenge in terms of biological response and biodegradability, making them less favourable in biomedical applications despite their possibility to be precisely tailored and thus achieve high reproducibility [31]. Natural polymers, on the other hand, are not cytotoxic and are mostly good candidates for cell adhesion and proliferation [32]. Several examples of naturally derived polymers used as the matrix material for MR hydrogels can be found in the literature, including proteins–gelatin [33,34,35], artificial polypeptides [36], polysaccharides–agarose [37,38], alginate [22,24,39,40,41], and carrageenan [38,42,43].

Hyaluronan (HA), a linear glycosaminoglycan (GAG) composed of D-glucuronic acid and N-acetyl-D-glucosamine linked by alternating β(1→3) and β(1→4) glycosidic bonds, is an excellent candidate for hydrogel formation. It is also the most abundant GAG occurring in the extracellular matrix (ECM). As such, it is largely responsible for the ECM’s physical properties and for modulating cellular behaviour [44]. Due to relatively available reactive functional groups (hydroxyl, carboxyl), HA is often chemically modified to allow hydrogel formation. Common approaches to cross-linking include methacrylation [45,46,47] or tyramine modification [48] followed by photocross-linking. Alternatively, thiol modification combined with enzymatic or chemical cross-linking [49,50,51] as well as amination [52] have been described.

However, the aforementioned modifications lead to non-reversible covalent cross-linking, which limits the possibility for further shape alterations. Dynamic cross-linking, such as Schiff base formation, is a convenient means to obtain pliable hydrogels exhibiting yield stress [53]. Such an approach, combined with photocross-linking, was used, e.g., by Wang et al., 2018 to obtain a non-cytotoxic hydrogel designed for scaffold fabrication [54].

HA- and iron-based particle composites present great potential in a wide variety of bioapplications. Numerous studies on magnetically responsive HA-based hydrogels can be found in the literature, describing materials suitable for magnetic heating [55,56], magnetic resonance imaging contrast agents [57], magnetomechanical neuromodulation [58], and controlled drug release [59]. In terms of MR hydrogels, Tran et al., 2021 reported collagen–HA hydrogels with added carbonyl iron particles (CIPs) to significantly alter the biological response through MRE [60]. Furthermore, it has been demonstrated that self-healing printable gels can show macroscopic and reversible change in dimensions in the presence of a magnetic field and thus may be potentially useful in 4D printing [61]. A similar phenomenon was observed in dynamically cross-linked self-healing ferrogel containing SPIONs [62]. Moreover, the presence of iron particles enhanced cell proliferation [61].

The literature review listed above presents several hypotheses for next steps in MR hydrogels for bioapplications:The use of dynamic cross-linking is more favourable compared to non-reversible covalent cross-linking, as it provides pliable materials that can exhibit yield stress [53,63]. Such behaviour is advantageous in terms of materials processing, namely injectability and extrudability, but also promises a possibility to achieve significant MRE [26,64].Iron-based magnetic fillers are non-toxic and biodegradable on account of biotransformation mechanisms [16,17]. The use of CIPs provides more-pronounced MRE compared to iron oxide microparticles or SPIONs due to higher saturation magnetization values [20].Purely HA-based hydrogel matrices are rather scarce in the literature even though they may find use in tissue engineering, scaffold fabrication, and on-demand cell stimulation through an external impulse [22].

Nevertheless, natural polymer-based magneto-responsive hydrogels are seldom reported in the literature. Furthermore, their use is mainly focused on magnetic heating [65] or magnetic guidance in targeted drug delivery systems [66] rather than MR-induced changes to hydrogel mechanics. Even when MRE is described in a natural-based hydrogel, the matrix does not purely utilize polysaccharides [60]. Therefore, our study can bring benefits to the field of magnetorheology by extending the utilizable materials to natural-based polymers, and thus find use in bioapplications.

In the present study, we describe a straightforward method for preparing a dynamically cross-linked MR hydrogel based on modified HA and CIPs. In order to obtain HA functionalised with adipic acid dihydrazide (ADH), traditional carbodiimide chemistry and 4-(4,6-Dimethoxy-1,3,5-triazin-2-yl)-4-methylmorpholinium chloride (DMTMM) mediated reactions are used. This ADH-modified HA is used as a precursor for Schiff base-linked hydrogels in combination with polysaccharide-based polyaldehydes. The dynamic character of cross-links provides a soft, shear-thinning matrix for preparation of MR hydrogel, which supposedly could lead to diminishing viscous restriction of the MR effect [26]. As the magnetically responsive filler, commercially available CIPs are used. The mechanical performance of the hydrogel is reversibly enhanced via an external magnetic field, as well as permanently by application of dual cross-linking. Finally, cytocompatibility is tested.

## 2. Results and Discussion

### 2.1. Polysaccharide Modification

Hydrogel preparation was preceded by two different biopolymer (HA and dextran) modifications. The first approach was based on covalent bonding of ADH to HA chains via two different activators of HA carboxyl groups. Initially, HA derivatization was carried out by the well-established carbodiimide-based activation of the HA carboxyl group using 1-ethyl-3-(3-dimethylaminopropyl)carbodiimide hydrochloride (EDC) and 1-hydroxybenzotriazole hydrate (HOBt) [67], as illustrated in Figure 1. Unfortunately, the reaction is strictly pH-dependent and also suffers from the creation of side products. Namely, bonding of N-acylurea to HA carboxyl groups makes the removal of the byproduct quite difficult [49]. Therefore, in order to synthesize ADH-modified HA (HA-ADH), an alternative coupling agent, DMTMM, was employed (Figure 1). Subsequently, the carboxyl group activators used, EDC and DMTMM, were compared and evaluated. Using either EDC or DMTMM, the desired product (HA-ADH) was successfully synthesized. Further, HA-ADH with different degrees of substitution (DS) was obtained depending on the activating agents. EDC provided HA-ADH with higher DS (22%) compared to DMTMM (DS 12%). Regarding reactions mediated by DMTMM, it was not possible to produce HA-ADH with higher DS because increasing reactant molar ratios led to undesirable in situ cross-linking of HA-ADH. On the other hand, the advantages of DMTMM include the reaction not being so strongly pH-dependent, and providing the product without side products, as was observed when using EDC. Despite the different DS, all prepared HA-ADHs were successfully employed for hydrogel preparation.

The second type of polysaccharide modification consisted of HA (Figure 2) and dextran oxidation. For this purpose, reactions mediated by NaIO${}_{4}$ were carried out [67,68], leading to polysaccharide polyaldehydes. Oxidized HA (HA-OX) and oxidized dextran (DEX-OX) were prepared with different degrees of oxidation (DO)—35 and 62 for HA-OX, and 49 for DEX-OX. The molecular weight of the polymers decreased to 7.5 kDa–9.0 kDa due to oxidation. The presence of formed carbonyl groups in biopolymer chains further enabled utilisation of HA-OX and DEX-OX for hydrogel formation after reaction with HA-ADH.

### 2.2. Schiff Base Formation-Induced Gelation

The hydrogels were obtained when HA-ADH was chemically cross-linked by polysaccharide polyaldehyes (Figure 3), resulting in Schiff base formation between aldehyde and amino groups present in polymer chains [69]. Table 1 gives the specifications of the hydrogel compositions. Hydrogel formation was evident by the significant increase in viscosity and mechanical stability over time. These changes in rheology were used to quantify the reaction rate by determination of gelation time.

In order to examine the effect of DO on gelation time, three polysaccharide polyaldehydes with different DO (i.e., HA-OX with DO 35 or 62 and DEX-OX with DO 49) were compared (Figure 4). The molecular weights of both HA-OX and DEX-OX were kept in the same range, so that DO coud be followed as an independent factor. It is evident from Figure 4A that in HA-OX, the gelation time directly corresponds to differences in DO, i.e., the number of reactive sites available. However, when comparing HA-OX with DO 62 to DEX-OX with DO 49, the shortest gelation time was observed in DEX-OX, despite its lower DO. Therefore, the gelation time is not dependent only on the number of aldehyde groups that participate in gelation, but may be also influenced by biopolymer properties such as polymer chain conformation. This quality is reflected in polymers’ hydrodynamic size, which can be characterized by the z-average diameters of the polymer coils in a solution [70]. Observed variation of the z-average diameter can be seen in Figure 4B. Based on this, it can be stated that the mean diameter of DEX-OX with DO 49 shows a very small value (4.36 ± 0.44 nm), and the resultant particle size distribution is narrow and monomodal, indicating the occurrence of one population of particles. On the other hand, the particle size distributions for both HA-OX samples are bimodal, which means the samples contain two different particle-size populations. HA-OX with DO 35 has a rather broad bimodal distribution that contains two particle-size fractions (2.80 ± 0.02 nm and 360 ± 50 nm). HA-OX with DO 62 has a rather narrow bimodal distribution that contain two size fractions (2.6 ± 0.4 nm and 280 ± 10 nm). This also confirms the generally accepted finding that HA in salt solutions has an expanded random-coil conformation, as would be expected for a flexible polyelectrolyte [71]. Moreover, the size of a polymer coil is proportional to its diffusion coefficient; thus, a smaller hydrodynamic radius benefits from fast diffusion through the sample [72]. Therefore, smaller particles of DEX-OX DO 49 move through the solvent more easily and accelerate the formation of Schiff bases [73]. The results indicate the complexity of chemical reactions among polymer chains and highlight the necessity of thorough evaluation of the macromolecules in order to accurately predict the outcomes.

### 2.3. Dual Cross-Linking of Hydrogels

A Schiff base as a dynamic bond can be easily depleted in some environments based on pH, ionic strength, and/or the presence of competitive agents capable of forming the same type of bonds. Thus, in order to achieve sufficient stability of the obtained hydrogels, ionic cross-linking with Fe${}^{3+}$ ions was additionally chosen as a simple and non-toxic option. Figure 5 shows a schematic of Schiff-base and dual cross-linked hydrogel preparation with the optional addition of a magnetoresponsive filler. Ionic bath treatment led to significant improvement of hydrogel structural integrity in cultivation conditions from less than 12 h to several days. We observed that hydrogels containing HA-OX (gels A, B, D, and E) were more stable (lasting 5 days in simulated cultivation conditions) compared to DEX-OX containing gels (C and F; lasting 2 days in simulated cultivation conditions) in the presence of Fe${}^{3+}$. Unlike dextran, HA contains free carboxyl groups in its structure, even in the oxidized form. As ionic cross-linking is fundamentally related to the reaction of multivalent ions and polar, typically carboxyl groups, it is reasonable to expect a more-pronounced effect in a system rich in the desired groups. A study conducted by Zellermann et al., 2013 [74] confirmed a significant effect of barium ions on behaviour of HA, while dextran, as an uncharged polysaccharide, was intact with the Ba${}^{2+}$ presence. Analogous behaviour can be expected in the presence of Fe${}^{3+}$ ions. This is also reflected in the possibility of using Fe${}^{3+}$ ions for obtaining HA-based hydrogels [75].

### 2.4. Rheology

#### 2.4.1. Schiff-Base Cross-Linked Hydrogels

Oscillation rheometry was chosen as the primary tool to evaluate hydrogel mechanics, as it can characterize steady shear flow viscoelastic behaviour as well as be approximated to steady-state characterization. The measurements reveal that HA-ADH with higher DS (gels A-C) leads to stronger hydrogel structures (storage modulus differs as much as three times compared to their respective gel D-F counterparts; see Figure 6A). A similar trend is observed for shear stress (Figure 7C,D). Furthermore, the dynamic character of the Schiff base linkages allows shear-thinning behaviour, as displayed in the decreasing viscosity with increasing shear rate (Figure 7A,B), which makes the hydrogels promising in terms of injectability and extrudability, qualities which are essential in medical applications and additive manufacturing [64,76]. The damping factor, on the hand, shows the opposite trend, confirming lower elasticity of hydrogels D–F (Figure 6B). It can be assumed that the lower DS achieved in DMTMM-mediated reactions leads to lower cross-linking density, thus weakening the structure.

Similar to gelation time, strength of hydrogels is not dependent solely on the DO of oxidized polysaccharides, although it is clearly a factor, as can be observed from comparison of storage moduli of hydrogels A and B and hydrogels D and E, as well as the generated shear stress. Nevertheless, the highest mechanical stability, reflected in storage modulus, viscosity, and shear stress (See Figure 6A and Figure 7), was found for hydrogels C and F containing DEX-OX with DO 49. Given the lower hydrodynamic radius of DEX-OX chains, as discussed earlier, it is possible to assume a more-compact polymer network was formed, which increased overall mechanical strength [77]. Thus, tuning of hydrogel rheological behaviour needs to account for the complex behaviour of polymer chains in order to successfully achieve desired properties.

#### 2.4.2. Dually Cross-Linked Hydrogels

Application of dual cross-linking leads to significant stiffening of the hydrogels. Shear stress increases as much as 10${}^{2}$ (Figure 7C,D), and storage modulus increases by a factor of 10${}^{3}$ (Figure 6A). This enhancement of mechanical stability allows easier manipulation of the compact hydrogel, and it increases endurance against mechanical stress, which is desirable for tissue analogue or replacement applications [78]. Nevertheless, the ionic cross-links lack the dynamic character of Schiff bases, which is documented by the milder slope of the viscosity–shear rate curve seen in Figure 7A,B. This confirms the necessity of subsequently applying the second cross-linking strategy in manufacturing in order to avoid material breakage while forming specific shapes [79]. Moreover, the damping factor increased in addition to the other rheological characteristics. Although it does not exceed one in any case (Figure 6B), proving that the hydrogels maintain their solid-like character, it also signals a rather significant increase in loss modulus, i.e., dissipated energy during mechanical stress. This unusual behaviour of hydrogels has been observed previously for polydimethylsiloxane hydrogels [80]. It may protect the components (particles, cells, etc.) enclosed in the hydrogel matrix during vibration or flow, and bring the overall behaviour closer to that of the native tissue.

### 2.5. Cytotoxicity

Cytotoxicity tests were performed on dually cross-linked HA-based hydrogels in order to establish the most suitable compositions for cell cultivation. The results can be found in Figure 8. All hydrogels based on EDC-modified HA were cytotoxic at 100% concentration. Presumably, this is the result of residual low-molecular-weight by-products from the modification reaction. Gel E was the only formulation that proved to be non-toxic in any concentration. It can be assumed that HA-OX with DO 35 gives very weakly bound hydrogels (gel A and D), which can easily deplete and thus release unreacted aldehyde groups, which are known for their cytotoxicity [81]. DEX-OX with DO 49 was less stable in cultivation medium than its HA-OX cross-linked counterparts. It appears that the reason for increased cytotoxicity is the same as in the previous case, i.e., the exposure of aldehyde groups. The lower stability of DEX-OX with DO 49 hydrogels may be associated with the inefficiency of additional stabilization mechanisms caused by the lower quantity of carboxyl groups available for ionic cross-linking due to the different structure of dextran compared to HA, as discussed in Section 3.3.

### 2.6. Swelling

Swelling ability is a core characteristic of hydrogels. It mainly depends on the material’s chemical nature, cross-linking density, solvent, and temperature [82]. Due to presumed utilization of the proposed hydrogels in bioaplications, the solvent and temperature were chosen to simulate biological conditions (PBS pH 7.4, 37 °C). We found that in such conditions, all of the hydrogel compositions reached equilibrium swelling within one hour. The differences are within measurement uncertainty; therefore, it appears that the individual specifics of the compositions have very little impact on swelling behaviour. Furthermore, all hydrogels except for gel D seemed stable for the whole time period (see Figure 9B). Gel D began to fall apart after two hours of swelling. This was likely due to a weak Schiff base network, which was insufficient even in the presence of secondary ionic cross-linking.

### 2.7. Mr Hydrogel Preparation

The tests conducted on hydrogels without CIPs were used as indicators to select materials best-suited for fabrication of scaffolds potentially utilizable in cell cultivation. In accordance with the aim of the study, only such materials were subjected to magneto-sensitising and characterisation of magneto-responsiveness. Hydrogels A–C were deemed unsuitable on account of cytotoxicity. Furthermore, the stability of gel D was not sufficient to be useful in cell cultivation applications (Figure 9). Despite the unfavourable cytotoxicity results of gel F, this hydrogel was used for further testing. Assuming the cytotoxicity was caused by the unstable part of the hydrogel, it would still be possible for cells to proliferate in the stable portion of the material. Therefore, two hydrogel compositions were chosen for MR experiments: gels E and F.

#### CIPs Characterization

Size, volume fraction, and magnetic properties are essential in understanding MR behaviour of the composite as a whole. Therefore, they need to be addressed individually as well as in the material. The used CIPs show saturation magnetization of 15.7 emu·g${}^{-1}$ and coercivity of 23 Oe (Figure 10A). Their magnetization is approximately 10 times lower and coercivity 20 times higher than that of carbonyl iron powder [83]. The most likely explanation is higher content of impurities, which are known to negatively affect magnetic properties of iron. Additionally, morphological analysis shows a spherical shape of the particles with a rather broad size distribution, as can be seen in Figure 10B,C.

### 2.8. CIP-Filled Hydrogel Magnetic Properties

CIPs, being hard inorganic particles, may induce stiffening of a hydrogel matrix due to their unyielding character. This effect is strongly connected to the particle volume fraction; specifically, non-linear growth begins at approximately 5 vol.% [84]. Due to high density of CIPs, it is possible to use significant weight fractions of particles as fillers, as their volume fraction is rather low. In the current study, we worked with CIPs at 30% weight fraction, thus reaching approximately 4% volume fraction. This amount of CIPs was chosen in order to obtain significant MRE while minimising the effects of hard filler particles. Additionally, it has been shown that higher filling of hydrogels (approx. 30 wt.%) diminishes shear-induced structural disintegration [85], which is desirable for injection and extrusion.

MRE is highly dependent on a material’s saturation magnetization. It has been found that dispersing CIPs in HA hydrogels leads to lowering of saturation magnetization compared to bare particles, as is apparent from Figure 11. Saturation magnetization was inversely proportional to hydrogel viscosity. It can be assumed that CIPs’ interparticle magnetic interactions are hindered in high-viscosity media. This also corresponds to the widely acknowledged inverse relationship between viscosity and MRE magnitude [15]. Coercivity—being dependent on the properties of a magnetic material, especially particle size [86]—remains unchanged, as the CIPs only serve as a filler and are not chemically changed during Schiff base formation.

### 2.9. Magnetorheology

#### 2.9.1. Schiff Base Cross-Linked MR Hydrogels

Using CIPs as a magneto-responsive filler induced several effects in hydrogel behaviour. Firstly, an interesting disproportion was found in the shear-flow behaviour without a magnetic field with respect to the polysaccharide polyaldehyde type. The presence of DEX-OX with DO 49 in gel F-CIP caused the storage modulus to increase with the addition of CIPs behaviour may be related to the formation of complexes between dextran and iron-containing particles [87]. Furthermore, the rapid gelation of DEX-OX with DO 49 would likely lead to more homogeneous distribution of CIPs, thus providing mechanical stiffening due to the content of hard particles [84]. This effect was not observed in gel E-CIP. Viscosity curves (Figure 12) confirm the shear-thinning character of hydrogels, as was described in the hydrogels without CIPs.

Damping factor shows the opposite trend, which corresponds to the expected shifting towards liquid-like or solid-like character of the respective hydrogels. In case of gel E-CIP, tan δ is close to 1, meaning that the viscous behaviour of the mixture is more pronounced than in gel E without CIPs (see Figure 13B). This may cause problems with manipulation in hydrogel applications. However, as long as sufficient viscosity is maintained, certain manufacturing techniques can be applied (e.g., casting or free-form fabrication) [64].

The main reason for incorporating magnetic filler is to induce magneto-sensitivity to the hydrogel, and thus obtain an MR material. In such a material, an external magnetic field induces (partial) alignment of CIPs. The filler thus creates chain-like clusters that provide internal mechanical support to the structure [20]. By comparing rheological characteristics—viscosity, shear stress, and storage modulus—it is clear that this effect is present in both of the examined hydrogel compositions. In order to quantify the magnitude of the MRE, we utilized the storage modulus in the magnetic field relative to an off-state value, and it is denoted as parameter Σ:(1)$$\Sigma =\frac{G_{H}^{{}^{\prime }}}{G_{0}^{{}^{\prime }}}$$
where G${}_{H}^{{}^{\prime }}$ represents the elastic modulus in a certain measured magnetic field, and G${}_{0}^{{}^{\prime }}$ represents the elastic modulus in the absence of a magnetic field. Expressing Σ as a function of magnetic field intensity H (Figure 13C) gives a clear comparison of MRE in various hydrogels, since the slope of the curve is proportional to the magnitude of magnetically induced strengthening. Σ is expressed in logarithmic scale for clarity. The examined CIPs-filled hydrogels show substantial strengthening in magnetic fields (10${}^{1}$–10${}^{3}$-times increase of storage modulus compared to the off-state value). The magnitude of the MR response is inversely proportional to hydrogel viscosity in the off state (Figure 12 and Figure 13C). Therefore, the MRE is more pronounced in a softer gel E-CIP, as highly viscous materials restrict the particles’ ability to move through their structure, thus preventing the formation of the chain clusters necessary for MRE [15].

Furthermore, cyclic switching of the magnetic field proved the response of all examined hydrogels to a magnetic field occurring within a few seconds and capable of fully recovering to their respective baseline states (Figure 13D). This offers an opportunity to control their mechanical properties in real time, and to potentially create a variable stimulus during cell cultivation [88].

#### 2.9.2. Dually Cross-Linked Hydrogels

Stabilization increased hydrogel stiffness significantly in the off-state (See Figure 14A) and hindered the viscous response, as is especially evident in gel E-CIP-Fe${}^{3+}$ (Figure 14B) as a consequence of additional cross-linking introduced by ionic bonds. Naturally, diminished the MR influence on the hydrogel’s mechanical properties (Figure 14C). Notably, the MR behaviour of gel F-CIP-Fe${}^{3+}$ stayed almost the same as before stabilization, while gel E-CIP-Fe${}^{3+}$ experienced a dramatic increase in stiffness as well as a significant decrease in MRE. These results further support the hypothesis of low efficiency of applied ionic stabilization in DEX-OX DO 49 hydrogel. Despite the decrease in MRE, a 25-times increase was induced in the 862 kA·m${}^{-1}$ magnetic field in gel E-CIP-Fe${}^{3+}$. This increase is sufficient to provide the desired response in cells [29]; thus, the hydrogels have potential for active cellular scaffold fabrication, e.g., in muscle regeneration. Myocytes positively react to dynamic mechanical stimuli due to their functional predispositions [89]. Additionally the absolute storage modulus of the hydrogels exceeded 10${}^{4}$ Pa, which is the reported strength of some some skeletal muscles [90]. Therefore, the presented materials may be particularly useful in this application.

Furthermore, the hydrogels retained the rapid responsiveness to magnetic field changes, which was observed in non-stabilized samples. The change in mechanical properties is instant, making the material suitable for fabrication of on-demand controlled scaffolds (see Figure 14D).

### 2.10. Porosity and Inner Morphology

SEM micrographs (Figure 15) of freeze-dried dually cross-linked gels with CIPs show porous structures in both samples. The average porosity is between 30–45%, and the average pore size is between 1000 μm^2^ and 2000 μm^2^. However, the variance of values is very broad, with the minimum value as low as 50 μm^2^, and the maximum reaching almost 1 mm^2^. There is no significant influence of oxidized polysaccharide characteristics on porosity or average pore size of hydrogels.

## 3. Materials and Methods

### 3.1. Chemicals

HA of the following molecular weights: 243 kDa, 1.18 MDa, and 1.5 MDa, was obtained from Contipro Inc., Dolni Dobrouc, Czech Republic.

Dextran from *Leuconstoc* spp. 70,000 g·mol${}^{-1}$ was obtained from Roth s.r.o., Trebarov, Czech Republic.

Ultrapure water (UPW) was prepared using a Milipore-Q system (Merck KGaA, Darmstadt, Germany.

Dimethyl sulfoxide (DMSO) ≥ 98% and NaHCO${}_{3}$, ACS reagent were obtained from VWR International, LLC., Radnor, PA, USA.

ADH ≥ 98% and phosphate buffered saline (PBS), 1X, pH 7.4, sterile were obtained from Thermo Fisher Scientific, Waltham, MA, USA.

EDC ≥ 99% was obtained from Roth s.r.o., Trebarov, Czech Republic.

HOBt ≥ 97%; DMTMM ≥ 96%, and NaIO${}_{4}$, ACS reagent for analysis were obtained from Merck KGaA, Darmstadt, Germany.

NaOH, p.a. was obtained from IPL, Uherský Brod, Czech Republic.

NaCl ≥ 99.5% and HCl 35% were obtained from Lach-ner, s.r.o., Neratovice, Czech Republic.

Fe${}_{3}$O${}_{4}$≥ 98% was obtained from PENTA s.r.o., Prague, Czech Republic.

A 3-(4,5-Dimethylthiazol-2-yl)-2,5-Diphenyltetrazolium bromide (MTT) cell proliferation assay kit was obtained from Duchefa Biochemie, Haarlem, The Netherlands.

Pentacarbonyl iron particles (i.e., CIPs), iron content over 97%, were obtained from BASF, Ludwigshafen, Germany.

### 3.2. Modification of HA

HA was grafted by ADH following two slightly modified respective reaction protocols utilizing either EDC [67] or DMTMM [91] as the HA carboxyl group activator.

#### 3.2.1. EDC-Mediated Reaction

First, HA (1 g, 2.5 mmol, 243 kDa) was dissolved in UPW in 50 °C while stirring overnight to obtain 0.3 wt.% solution. When the HA solution was cooled to 25 °C, ADH (13 g, 75 mmol) was added, followed by pH adjustment to 6.8 with 0.1 M NaOH solution. Then, EDC (1.9 g, 10 mmol) and HoBt (1.4 g, 10 mmol) dissolved in 6 mL of DMSO/UPW (1/1, *v*/*v*) were added to the reaction mixture. The pH of the reaction decreased gradually and had to be continuously adjusted to 6.8 during the first hour of the reaction. After that, the reaction proceeded under stirring at 25 °C for 22 h. The reaction mixture was then transferred into dialysis tubes (cut-off 12,000 Da) and dialysed against salt solution (12.5 g NaHCO${}_{3}$ and 12.5 g NaCl per 10L) for one day, then against UPW for two days. ADH-modified HA (HA-ADH) was obtained after freeze-drying to yield 1.0 g (94%) with DS 22%. ${}^{1}$H NMR (D${}_{2}$O, 400 MHz) δ: 4.53–4.44 (2H, HA anomeric, CH), 3.82–3.32 (10H, HA skeletal, CH), 2.88 (4H, urea, NCH${}_{2}$), 2.37 (2H, ADH, CH${}_{2}$NHNHCO), 2.23 (2H, ADH, CH${}_{2}$NHNH${}_{2}$), 2.00 (3H, HA, NHCOCH${}_{3}$), 1.65 (4H, ADH, CH${}_{2}$CH${}_{2}$) ppm.

#### 3.2.2. DMTMM-Mediated Reaction

The 1% HA (5 g, 12.5 mmol, 243 kDa) solution in UPW was prepared in the same manner as described previously. Initially, ADH dissolved in 4.5 mL UPW was added to this HA solution. The pH of the reaction mixture was adjusted to 6.5 using HCl or NaOH solution. After that, DMTMM (865 mg, 3.1 mmol) was added, and the reaction was left to proceed for 24 h under stirring at 25 °C. Purification and product isolation followed the same manner as the previous case. HA-ADH with DS 12% and a yield of 2.9 g (53%) was obtained. ${}^{1}$H NMR (D${}_{2}$O, 400 MHz) δ: 4.52–4.43 (2H, HA anomeric, CH), 3.82–3.31 (10H, HA skeletal, CH), 2.37 (2H, ADH, CH${}_{2}$NHNHCO), 2.23 (2H, ADH, CH${}_{2}$NHNH${}_{2}$), 2.00 (3H, HA, NHCOCH${}_{3}$), 1.65 (4H, ADH, CH${}_{2}$CH${}_{2}$) ppm.

### 3.3. Oxidation of HA and Dextran

Polysaccharide polyaldehydes were obtained by periodate oxidation according to published procedures [67,68]. Specifically, the respective polysaccharide was dissolved in UPW overnight at 50 °C under stirring. Afterwards, the appropriate amount of sodium periodate dissolved in water was added to the polysaccharide solution under vigorous stirring at 25 °C. The reactions were carried out in the dark at 25 °C for a given time (Table 2) under stirring. The reaction mixtures were then placed into dialysis tubes (cut-off 3000 Da) and dialysed against UPW for three days. After that, the purified product was obtained by freeze-drying (Table 3).

### 3.4. Schiff Base Linkage Formation

Hydrogel precursors, i.e., HA-ADH and polysaccharide polyaldehyde, were dissolved separately in PBS overnight at room temperature to obtain 2 wt.% solutions. Then, the solutions were mixed in a 1:1 volume ratio, and gelation occurred spontaneously. This procedure yielded Schiff-base cross-linked hydrogels.

For MR hydrogel preparation, 60 wt./vol.% of CIPs was added to the polysaccharide polyaldehyde solution and vortexed thoroughly to obtain homogeneous dispersion. Due to the fast sedimentation of CIPs, it was necessary to ensure immediate mixing with HA-ADH solution to induce gelation via Schiff base formation.

### 3.5. Dual Cross-Linking of Hydrogels

The covalently cross-linked hydrogels with and without CIPs were submerged in a 2 wt.% aqueous solution of FeCl${}_{3}$ for 1 h at 25 °C. After removal from the Fe${}^{3+}$ bath, the samples were placed in a large amount of UPW for three days at 25 °C, with the water changed three times a day. To confirm that the excessive Fe${}^{3+}$ had been washed out, conductivity of the water was measured before each change. This procedure was unchanged regardless of CIP presence.

The stability of dually cross-linked hydrogels during cell cultivation was established in simulated cultivation conditions, i.e., shaking while submerged in a cultivation medium at 37 °C. The state of the hydrogel samples was checked daily, and the cultivation medium was changed every day as well.

### 3.6. Determining DO

The number of aldehyde groups per 100 saccharide subunits, i.e., the DO, was established using the hydroxylamine hydrochloride method [92]. The measurements were performed using an automatic titrator T50 (Metler Tolledo, Greifensee, Switzerland).

### 3.7. Nuclear Magnetic Resonance

Proton nuclear magnetic resonance (${}^{1}$H NMR) spectra were recorded on a JEOL ECZ 400 (JEOL Ltd., Tokyo, Japan) operating at a ${}^{1}$H frequency of 399.78 MHz at 60 °C. The samples were dissolved in D${}_{2}$O at concentrations of 10 mg·mL${}^{-1}$ for the analysis. Water was used as a reference signal and was set at 4.75 ppm. The DS of HA-ADH was calculated by comparing a molar ratio (integral) of the peaks assigned to an HA N-acetyl group (3H) at 2.00 ppm and two methylene groups of ADH (4H) at 1.65 ppm.

### 3.8. Molecular Weight Determination

The average molecular weight and distribution curve of oxidized polysaccharides (HA, dextran) were determined by gel permeation chromatography performed on a high-performance liquid chromatograph system equipped with a refractive index (RI) detector (Shimadzu Prominence, LC-20 series, Shimadzu Corporation, Kyoto, Japan) with the following parameters: 0.1M PBS solution at 7.4 pH, flow 0.8 mL·min${}^{-1}$, oven temperature 30 °C, columns PL aquagel-OH 60 8 μm, 300 × 7.5 mm and PL aquagel-OH 40, 8 μm, 300 × 7.5 mm were connected in series. Pullulan standards were used for molecular weight calibration; analyses were based on RI data.

### 3.9. Dynamic Light-Scattering Measurement

Hydrodynamic diameters of HA-OX and DEX-OX solutions were determined by dynamic light scattering on a Zetasizer Nano ZS90 (Malvern Instruments, Malvern, UK). The results were expressed as intensity-weighted z-average diameter. Measurements of all diluted samples (concentration 2 wt.%) were carried out at a scattering angle of 173° with a 4 mW He–Ne laser operating at 633 nm and 25 °C. Before the measurements, all samples were filtered through a 0.45 μm pore-size polytetrafluoro ethylene syringe filter (Millipore, UK). Each sample was measured in triplicate.

### 3.10. Particle Size Analysis

Particle size analysis of CIPs was done on a laser diffraction particle sizing instrument (Mastersizer 3000, Malvern Instruments Ltd., Malvern, UK). The results were expressed as volume distribution. The laser diffractive method provides the standard percentile readings Dv 50, Dv 10, and Dv 90. Each sample was measured five times, and the mean and standard deviation were subsequently calculated.

### 3.11. Magnetometry

The bare CIPs and the magneto-responsive hydrogels were characterized according to their magnetization with a vibrating sample magnetometer Lake Shore 7404 (Lake Shore Cryotronics Inc., Westerville, OH, USA) at room temperature in room air in a magnetic field of up to 10 kOe. The amplitude and the frequency of the vibration were 1.5 mm and 82 Hz, respectively.

### 3.12. Rheology and Magnetorheology

All measurements were performed using a rotational rheometer, Anton-Paar MCR 502 (Anton Paar, Graz, Austria) at 25 °C under normal pressure in room air.

Time dependence of fresh polymer solution (2 mL) viscosity was measured using a double-gap measuring system DG26.7/T200/SS oscillating at constant 10% deformation with a constant angular frequency of 10 rad·s${}^{-1}$. Changes in rheological behaviour were followed over a 14 h time sweep. The gelation point was defined as the crossover between storage and loss moduli. It ought to be noted that sample preparation caused an approximately 1 min delay between the reaction start and obtainment of the first datum.

On the other hand, the rheology of Schiff base and dual cross-linked hydrogels was performed using a Peltier measuring system P-PTD200/62/TG with PP/25 in oscillation at constant 1% deformation with angular frequency sweep increasing from 0.1 to 100 rad·s${}^{-1}$. The hydrogel samples were prepared 12 h in advance for Schiff-base cross-linking, or 4 days in advance for dual cross-linking in order to satisfy the necessity to wash out excessive Fe${}^{3+}$ ions. The samples were in the form of circular plates with a diameter of 30 mm and a thickness of 2 mm.

Furthermore, CIP-filled hydrogels were subjected to magnetorheological measurement in a homogeneous magnetic field. For magnetorheological investigations, an MRD 180/1T measuring cell with PP 20/MRD/TI measuring geometry, and a Julabo temperating unit ensured that all investigation were performed at room temperature. The parallel-plate measuring system diameter was 20 mm. The magnetic field strength was consecutively set to 0 kA·m${}^{-1}$, 432 kA·m${}^{-1}$, and 862 kA·m${}^{-1}$. Additionally, a time sweep measurement while periodically switching the magnetic field between on and off states with an interval of 30 s was done.

Viscosity and shear strain are shown with respect to shear rate in the measured range, i.e., from 0.001 s${}^{-1}$ to 1 s${}^{-1}$. Storage modulus and damping factor, on the hand, are expressed as the mean value measured in the range 0.01 s${}^{-1}$ to 0.1 s${}^{-1}$.

### 3.13. Morphological Analysis

CIP morphology and the inner morphology of the material was observed using scanning electron microscope (SEM) imaging of freeze-dried samples in vertical sections using a Phenom Pro (Thermo Fisher Scientific, Waltham, MA, USA) at an accelerating voltage of 5 kV. The samples were sputtered with a gold–palladium layer prior to imaging. Image processing was performed with the aid of ImageJ software.

### 3.14. Swelling

Gravimetrical measurement protocol was used to determine the swelling behaviour of dually cross-linked hydrogels. Specifically, freeze-dried samples of known mass were immersed in PBS (0.1 M, pH 7.4). The equilibrium buffer uptake, S(e) (%), of hydrogels was determined by weighing the swollen samples at selected time intervals of 1, 2, 6, 15, 30, 60, 120, 240, 360 and 1440 min. The presented results are the average of 3 measurements. The samples were maintained at 37 °C throughout the measurement. The buffer and temperature were chosen with respect to biological testing requirements.

### 3.15. Cytotoxicity Testing

Cytotoxicity testing was done according to ISO standard 10993 using the NIH/3 T3 cell line. Sterilization was done prior to cytotoxicity testing by shaking samples in 70% ethanol for 1 h at laboratory temperature. Elution of the ethanol from samples was done by aspiration, a short rinsing in PBS, and then by shaking the samples in UPW for 48 h at laboratory temperature. After UPW aspiration, extracts were created. Extracts were prepared according to ISO standard 10993-12 (100 mg of hydrogel/1 mL of media). The tested material was incubated in a cultivation medium for 24 h at 37 °C with stirring. The parent extracts (100%) were then diluted in medium to obtain a series of dilutions with concentrations of 75 vol.% and 50 vol.%. All extracts were used for up to 24 h. The cells were seeded at a concentration of 10${}^{5}$ per well in 96-well plates. After the pre/incubation period (24 h), the extracts were filtered using a syringe filter with membrane pore size of 0.22 μm (TPP, Trasadingen, Switzerland) in order to ensure that no residual hydrogel was present. The filtered extracts in required dilutions were added to the cells and incubated for 24 h. Subsequently, tetrazolium salt was used to determine cell viability. Absorbance was measured using a microplate reader, Infinite M200 PRO (Tecan, Männedorf, Switzerland) at 570 nm, and the reference wavelength was adjusted to 690 nm. The results are presented as the percent reduction of cell viability when compared to cells cultivated in medium without the extracts of tested materials. Morphology of cells from the culture plates was observed using an inverted Olympus phase-contrast microscope (IX 81).

## 4. Conclusions

In this research paper, we have described synthesis of HA-based polysaccharide hydrogels utilizing dual cross-linking. Furthermore, the hydrogels were given magneto-responsiveness through incorporating CIPs in their structure in a simple and straightforward manner, maintaining low toxicity of the production process. In order to produce hydrogels, a reaction between HA-ADH and polysaccharide polyaldehydes yielding Schiff-base linkages was used. EDC and DMTMM carboxyl group activating agent respectively were employed for HA-ADH preparation, and the products were compared and evaluated. Despite DMTMM providing a product with lower DS, we proved that the DMTMM-mediated reaction led to pure HA-ADH without by-products. Moreover, product purity had a positive effect on cytocompatibility of resulting hydrogels compared to EDC. Therefore, it may be more suitable for application in cell cultivation.

We found that even though the DO of oxidized polysaccharides is a non-negligible factor in hydrogel mechanical performance, it is necessary to consider the specifics of polysaccharide chains, such as the hydrodynamic radius. The small size of DEX-OX with DO 49 polymer coils caused rapid gelation (less than 30 min) compared to HA-OX with DO 62, which, despite the higher quantity of available reactive sites, reached gelation in 50 min or 3 h depending on the DS of HA-ADH. Similarly, a disproportion in storage modulus was observed, as DEX-OX hydrogels were twice as strong as HA-OX with DO 62 due to more-compact structure provided by the lower hydrodynamic radius of its polymer coils.

Ionic cross-linking additionally increased the storage modulus of presented hydrogels as much as 10${}^{2}$ times compared to sole Schiff base cross-linking. The effect of Fe${}^{3+}$ ions was more pronounced in hydrogels containing only HA in the polymer matrix. The use of CIPs as magneto-responsive filler substantially increased hydrogel mechanical performance in an external magnetic field. The magnitude of the MRE was inversely proportional to hydrogel off-state viscosity. Overall storage modulus of the hydrogels reached 10${}^{4}$ Pa, and the maximum change induced by MRE was 10${}^{3}$ higher than the off-state value of Schiff-base cross-linked hydrogels, and 25 times the value without the magnetic field for dually cross-linked hydrogels. This substantial increase promises opportunities for mechanical stimulation of cells such as myocytes.

To summarize, the current work demonstrates successful preparation of HA-based MR hydrogels utilizing CIPs as an active filler. To the best of our knowledge, this is the first time an HA-based material has been used as a matrix for CIPs to secure rapid and reversible stiffening through MRE. Their cytocompatibility encourages further research of these hydrogels in terms of tissue engineering, especially in skeletal muscle regeneration applications. These qualities promise the presented materials to be useful in biological applications, specifically scaffold fabrication as injectable or 3D printable hydrogels.

## Figures and Tables

**Figure 1 ijms-23-09633-f001:**
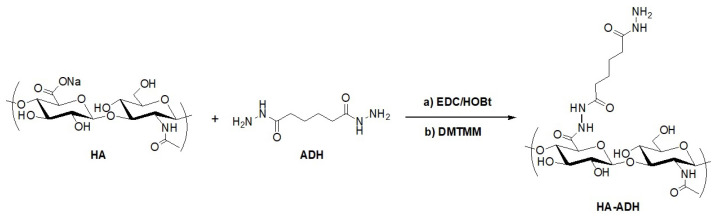
Synthesis of HA modified with ADH using (**a**) EDC/HOBt and (**b**) DMTMM.

**Figure 2 ijms-23-09633-f002:**
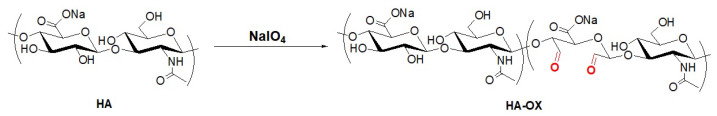
Oxidation of HA.

**Figure 3 ijms-23-09633-f003:**
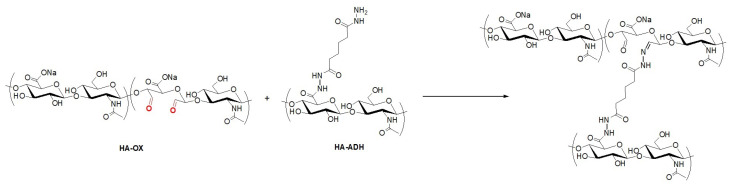
Schiff base formation between HA-OX and HA-ADH.

**Figure 4 ijms-23-09633-f004:**
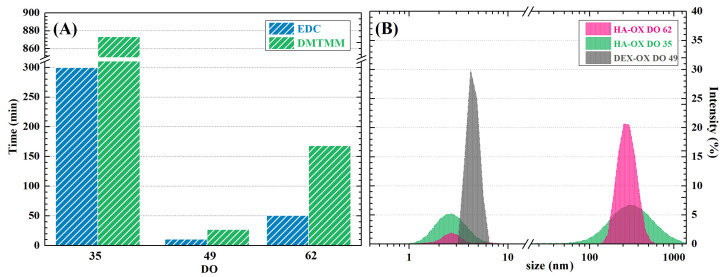
(**A**) Gelation times of HA gel formulations; (**B**) Hydrodynamic radius of oxidized polysaccharides.

**Figure 5 ijms-23-09633-f005:**
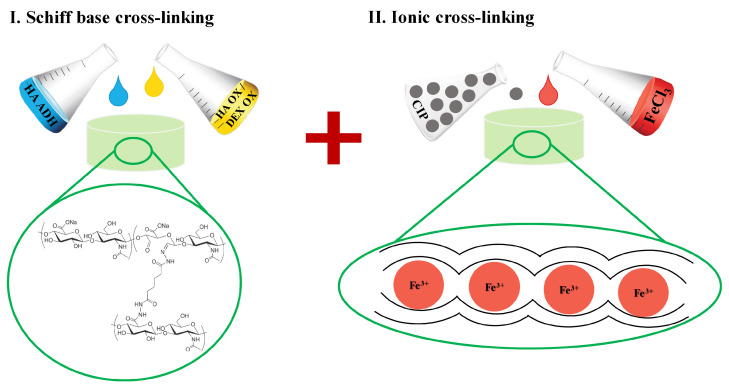
Schematic representation of (**I**) Schiff base cross-linking and (**II**) subsequent ionic cross-linking of prepared hydrogels with optional incorporation of magnetoresponsive filler CIPs.

**Figure 6 ijms-23-09633-f006:**
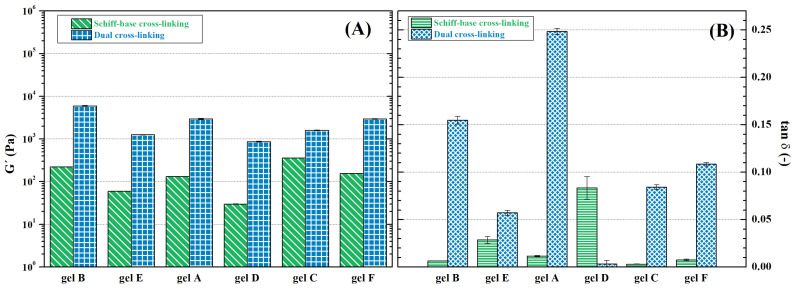
Comparison of (**A**) storage modulus and (**B**) damping factor of hydrogels cross-linked via Schiff bases and dually.

**Figure 7 ijms-23-09633-f007:**
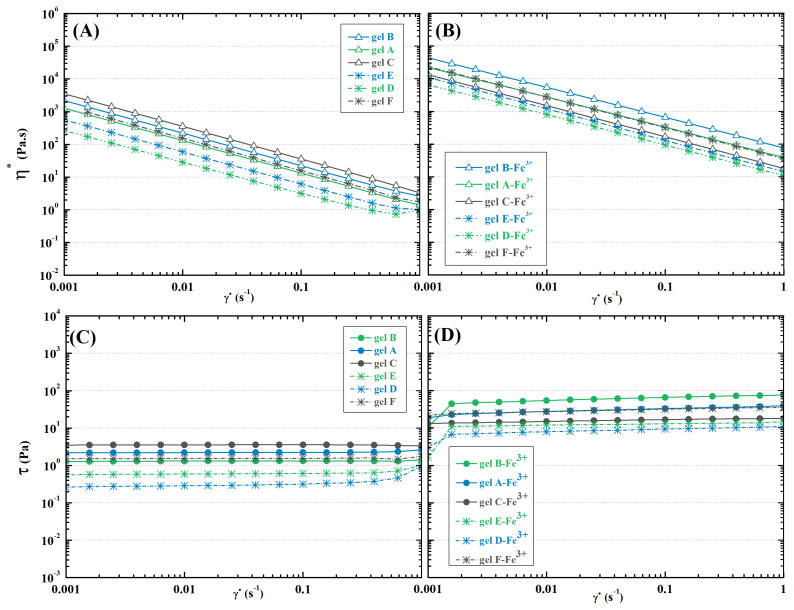
Rheological characterisation of the hydrogels: (**A**) Complex viscosity–shear rate curves of Schiff base cross-linked and (**B**) dually cross-linked hydrogels; (**C**) Shear stress–shear rate curves of Schiff base cross-linked and (**D**) dually cross-linked hydrogels.

**Figure 8 ijms-23-09633-f008:**
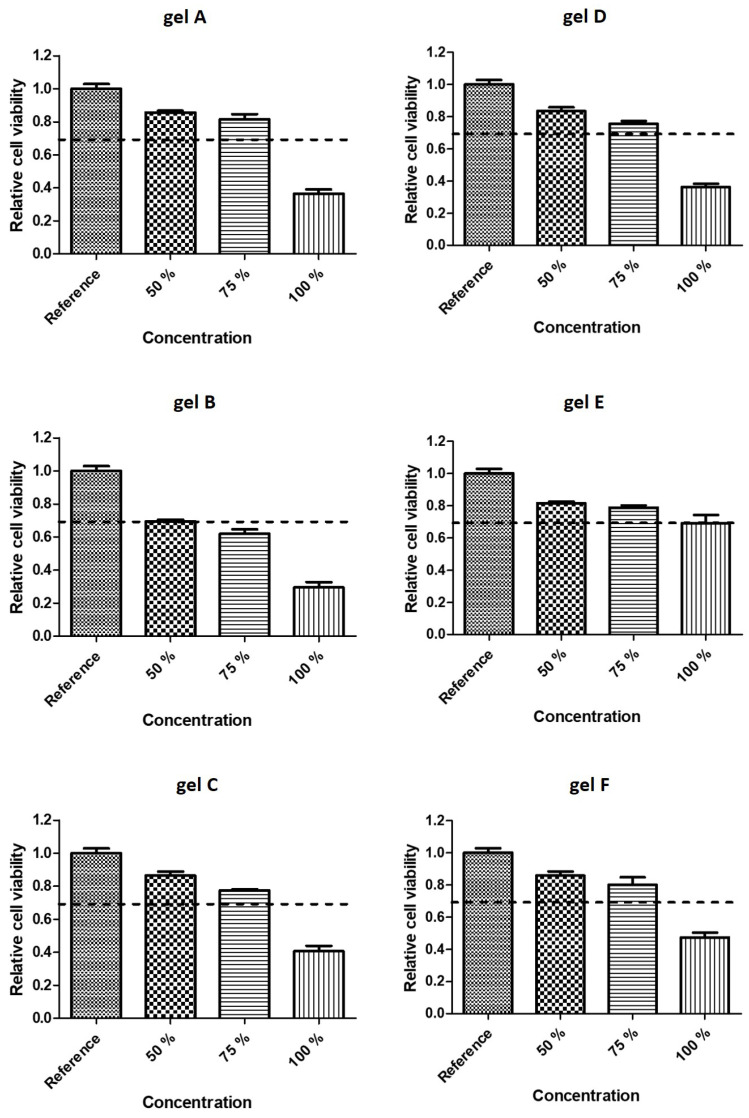
Cytotoxicity of dually cross-linked hydrogels.

**Figure 9 ijms-23-09633-f009:**
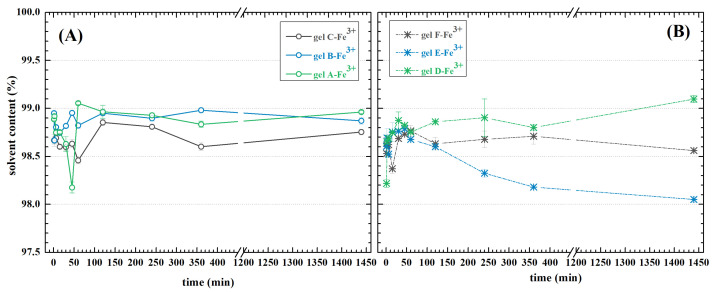
Swelling of dually cross-linked hydrogels. (**A**) Hydrogels based on HA-ADH modified using EDC; (**B**) Hydrogels based on HA-ADH modified using DMTMM.

**Figure 10 ijms-23-09633-f010:**
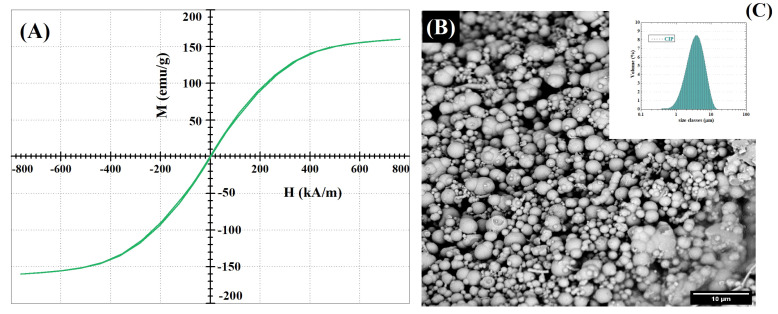
Characterization of CIPs used in MR hydrogels: (**A**) Magnetization hysteresis curve of CIPs; (**B**) SEM micrograph of CIPs; (**C**) CIP size distribution.

**Figure 11 ijms-23-09633-f011:**
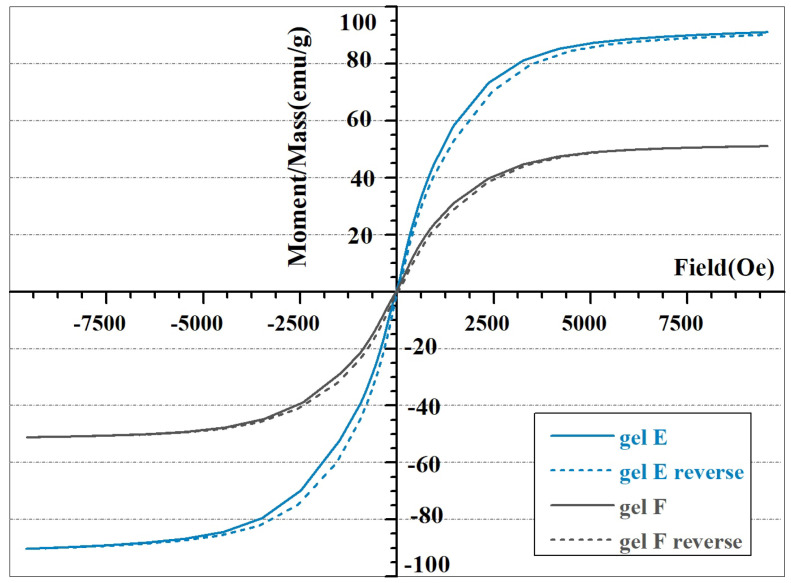
Magnetization hysteresis curves of Schiff base cross-linked HA gels containing 30% CIPs.

**Figure 12 ijms-23-09633-f012:**
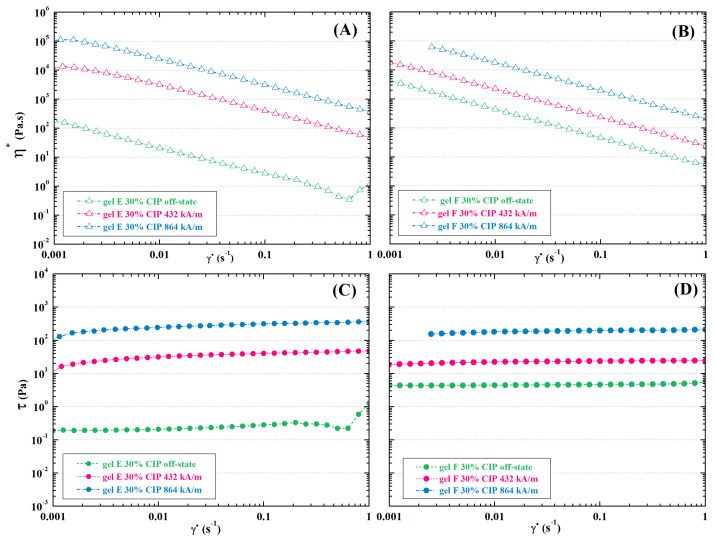
Shear-flow characterisation of CIP-filled Schiff base cross-linked gels: (**A**,**B**) Complex viscosity –shear rate curves and (**C**,**D**) shear stress–shear rate curves of gel E-CIP (**A**,**C**) and gel F-CIP (**B**,**D**) in an external magnetic field.

**Figure 13 ijms-23-09633-f013:**
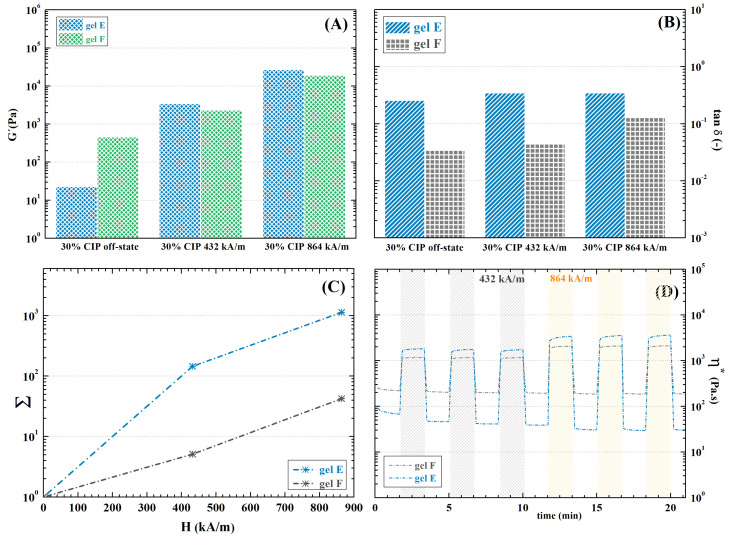
Mechanical characteristics obtained from rheological measurements of CIP-filled Schiff-base cross-linked hydrogels in an external magnetic field: (**A**) storage modulus; (**B**) damping factor; (**C**) intensity of MRE expressed as increase in storage modulus related to magnetic field intensity; (**D**) cyclic magnetic field exposure.

**Figure 14 ijms-23-09633-f014:**
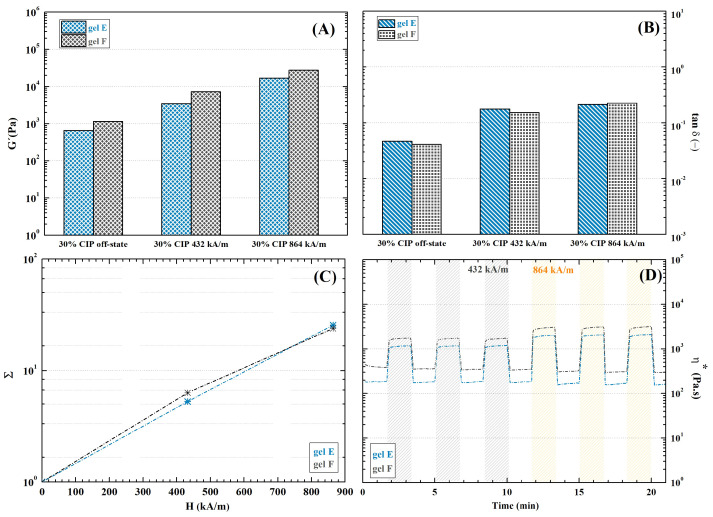
Mechanical characteristics obtained from rheological measurements of CIP-filled dually cross-linked hydrogels in an external magnetic field: (**A**) storage modulus; (**B**) damping factor; (**C**) intensity of MRE expressed as increase in storage modulus related to magnetic field intensity; (**D**) cyclic magnetic field exposure.

**Figure 15 ijms-23-09633-f015:**
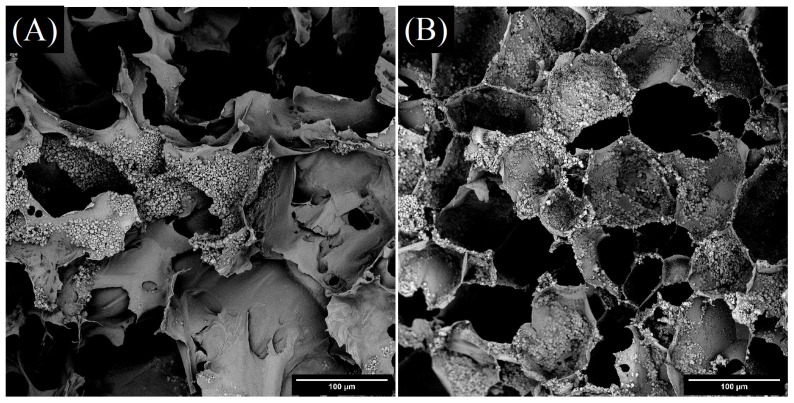
SEM micrographs of freeze-dried CIP-filled hydrogels: (**A**) gel E; (**B**) gel F.

**Table 1 ijms-23-09633-t001:** List of prepared hydrogel samples and their compositions as indicated by the checkmarks at respective components.

Sample Name	HA-ADH (EDC)	HA-ADH (DMTMM)	HA-OX DO 35	HA-OX DO 62	DEX-OX DO 49
gel A	✓		✓		
gel B	✓			✓	
gel C	✓				✓
gel D		✓	✓		
gel E		✓		✓	
gel F		✓			✓

**Table 2 ijms-23-09633-t002:** Specification of polysaccharide oxidation conditions.

	Initial Molecular Weight (kDa)	Mass of HA (g)	Molar Amount of HA (mmol)	Weight Fraction of HA (wt.%)	Mass of NaIO_4_ (g)	Molar Amount of NaIO_4_ (mmol)	Time (hours)
HA-OX A	1500	1.5	3.8	1	0.88	4.1	10
HA-OX B	1180	1.5	3.8	1	0.88	4.1	10
DEX-OX	70	3	18.5	13	1.58	7.4	4

**Table 3 ijms-23-09633-t003:** Characterisation of oxidized polysaccharides.

	Yield (g)	Yield (%)	DO	Final Molecular Weigth (kDa)
HA-OX A	0.55	36	62	7.6
HA-OX B	0.78	51	35	8.4
DEX-OX	2.66	89	49	8.8

## Data Availability

Not applicable.

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
