# Peer review of "Formulation of Magneto-Responsive Hydrogels from Dually Cross-Linked Polysaccharides: Synthesis, Tuning and Evaluation of Rheological Properties"

_ijms, 2022, doi:10.3390/ijms23179633_

Round 1

Reviewer 1 Report

The title must be changed by adding words like formulation, synthesis, evaluation etc. The abstract needs to be improved by adding some quantitative results. The overall conclusion in the abstract can be improved by presenting specific practical approach. The introduction is good but must be presented with detailed background of the hydrogels with proper practical examples like hydrogels for plastic industry, hydrogels for drug delivery, hydrogels for suiters. How it is important for future if applied? Some experiment of the practical analysis should be added as mentioned above; like if it is biodegradable, then add drug delivery, drug release, drug release mechanism etc. Results portion is sufficient and in accordance to the methods. The discussion seems a bit poor, which may be improved. 

Reviewer 2 Report

This manuscript describes a simple method for the preparation of a dynamic cross-linked hydrogel based on modified Hyaluronan and carbonyl iron particles. Traditional carbodiimide chemistry and 4-(4,6-dimethoxy-1,3,5-triazin-2-yl)-4-methylmorpholinium chloride mediated reactions were used to obtain Hyaluronan functionalized with adipic acid dihydrazide. The mechanical characteristics as well as the cytocompatibility of the obtained hydrogels were evaluated. The experiments in this study are well planned and of good technical quality. However, several issues need to be addressed for the manuscript to be published in International Journal of Molecular Sciences. To improve the manuscript I suggest the following comments:

- a schematic representation for the preparation of hydrogels should be introduced

- the morphology of the obtained hydrogels is not discussed at all. Modification of the structure according to the composition must be evaluated. A TEM analysis should also be performed to visualize the CIP distribution in the hydrogel structure.

- for cytotoxicity tests the authors used the extract method. When expressing the results the concentration as mg or ug hydrogel/ml culture medium should be taken into account.

Round 2

Reviewer 2 Report

The authors considered only part of the reviewers' recommendations.
Morphological characterization of hydrogels should appear in the manuscript, not in the supplementary material.
In figure 8, the concentration must be presented as mg or ug hydrogel/ml culture medium. The authors used an extract from 100 mg of hydrogel in 1 ml of culture medium. But when the extract was added over the cells, isn't there another amount of medium that must be taken into account?
